# Synthesis and Antioxidative Properties of 1,2,3,4-Tetrahydropyridine Derivatives with Different Substituents in 4-Position

**DOI:** 10.3390/molecules27217423

**Published:** 2022-11-01

**Authors:** Daniele Aiello, Hendrik Jonas, Anna Carbone, Daniela Carbone, Camilla Pecoraro, Luisa Tesoriere, Jens Köhler, Bernhard Wünsch, Patrizia Diana

**Affiliations:** 1Dipartimento di Scienze e Tecnologie Biologiche Chimiche e Farmaceutiche (STEBICEF), Università degli Studi di Palermo, Via Archirafi 32, 90123 Palermo, Italy; 2Dipartimento di Scienze della Vita, Università degli Studi di Modena e Reggio Emilia, Via Campi 103, 41125 Modena, Italy; 3Institut für Pharmazeutische und Medizinische Chemie, Westfälische Wilhelms-Universität Münster, Corrensstraße 48, D-48149 Münster, Germany; 4Dipartimento di Farmacia, Università degli Studi di Genova, Viale Benedetto XV 3, 16132 Genova, Italy

**Keywords:** indicaxanthins, betalamic acid, antioxidant, dehydrobromination, TEMPO oxidation, (E)-(Z) configuration, piperidin-4-ones, *cis*/*trans* diastereomers, *Wittig* reaction, *Lemieux*–*Johnson* oxidation, Folin–Ciocalteu

## Abstract

Natural products are an excellent source of inspiration for the development of new drugs. Among them, betalains have been extensively studied for their antioxidant properties and potential application as natural food dyes. Herein, we describe the seven-step synthesis of new betalamic acid analogs without carboxy groups in the 2- and 6-position with an overall yield of ~70%. The Folin–Ciocalteu assay was used to determine the antioxidant properties of protected intermediate **21**. Additionally, the five-step synthesis of betalamic acid analog **35** with three ester moieties was performed. Using NMR techniques, the stability of the obtained compounds towards oxygen was analyzed.

## 1. Introduction

Natural colors in fruits and plants are essential for photosynthesis, pollination, and seed dissemination [1,2]. Colors in plants are caused by three chemically distinct pigment types, anthocyanins (**1**), betalains (**2**), and carotenoids (**3**) (Figure 1). Anthocyanins are water-soluble pigments that give blue, red, and purple hues. The chemistry, production, and distribution of these compounds have been extensively investigated in the past [3,4,5,6,7].

In the last fifty years, there has been a growing interest in betalains. With few exceptions, plants and fruits of the order Caryophyllales exhibit a range of colors from red/purple to orange/yellow, due to the presence of these hydrophilic pigments. Initially, betalains were classified as anthocyanins. However, it was later discovered that the main enzymes required for the formation of anthocyanins are not present in betalain-producing plants [8,9].

Betalains are nitrogen-containing water-soluble pigments. Their biosynthesis in plants starts with L-tyrosine (**4**), which is converted into _L_-3,4-dihydroxyphenylalanine _L_-DOPA (**5**). The enzyme tyrosinase was thought to be responsible for the hydroxylation of _L_-tyrosine [10]. Recently, it has been observed that cytochrome P-450 monooxygenases are also able to catalyze this reaction [11]. Through the action of the enzyme 4,5-DOPA-extradiol dioxygenase, _L_-DOPA is converted into 4,5-*seco-*DOPA (**7**). Spontaneous cyclization of 4,5-*seco*-DOPA leads to betalamic acid (**9**), the key intermediate in the biosynthesis of all betalains. Moreover, tyrosinase is also involved in the oxidation of _L_-DOPA to *o*-DOPA-quinone (**6**), which undergoes cyclization to form cyclo-DOPA (**8**). Spontaneous condensation between cyclo-DOPA (**8**) and betalamic acid (**9**) leads to red and violet pigments named betacyanins (e.g., betanidin (**11**)). Alternatively, the reaction of betalamic acid (**9**) with amino acids or amino acid derivatives provides yellow-colored betaxanthins (e.g., indicaxanthin (**12**)) (Figure 2).

Betaxanthins are yellow, regardless of the amino acid or amine condensed with betalamic acid. Betaxanthins have a maximum absorption wavelength of 480 nm, while betacyanins show a maximum absorption wavelength of 536 nm. Additionally, a sugar moiety is linked to one of the phenolic OH moieties in betacyanin’s cyclo-DOPA portion [10,12,13].

The food industry has demonstrated a growing interest in these pigments as food colorants [14,15]. Moreover, betalains exhibit antioxidant and radical scavenging activity [16,17,18,19,20,21,22,23,24,25]. Numerous investigations have demonstrated that indicaxanthin (**12**) possesses both antiproliferative and chemoprotective properties [26,27,28].

The majority of betalains employed in biological research are extracted directly from plants by solid–liquid extraction. Maceration of vegetables facilitates the diffusion of the substances. Additional cellular components are released after tissue breakdown, which makes further purification necessary. Although betalains are typically extracted with H_2_O, other solvents such as MeOH and EtOH are frequently added to aid the extraction process. Unfortunately, this approach requires longer extraction time, additional purification procedures, and provides limited yields. As a result, innovative extraction methods were used to increase the efficiency of the isolation process of betalains, such as diffusion extraction, ultrafiltration, reverse osmosis, and cryogenic freezing [25,29,30,31,32]. Another significant issue encountered during the extraction and purification of these pigments is their chemical instability when exposed to oxygen, acids, bases, light, and heat. These parameters have a considerable impact on the extraction and purification procedures’ efficiency [33]. Several strategies for increasing the stability of betalains have been implemented, most notably in the food industry [34].

Betalamic acid (**9**) is a critical intermediary in the formation of both kinds of betalains. Although two syntheses of this compound have already been reported [35,36,37], the first synthesis developed by Dreiding et al. [35,37] started with chelidamic acid (**I**)**.** Hydrogenation of **19** in the presence of rhodium on activated alumina afforded an all-*cis-*configured piperidine derivative, which was converted into the dimethyl ester **II** upon treatment with methanol and HCl. Oxidation of the secondary alcohol led to the formation of piperidin-4-one **III**. To avoid overoxidation to the corresponding pyridine derivative, a polymeric carbodiimide was used for the Pfitzner–Moffatt oxidation and the transformation was carefully monitored. For the introduction of the side chain, a fully methyl-protected semicarbazide was employed as the Horner–Wittig reagent. This reagent led to the formation of hydrazone **IV** as a pure €-configured diastereomer (C=N bond). Dehydrogenation of **IV** with *t*-butyl hypochlorite and triethylamine (NEt_3_) provided dihydropyridine **V** as a 7:3 mixture of (*E*)- and (*Z*)-configured diastereomers. In this case, (*E*) and (*Z*) configuration refers to the exocyclic C=C double bond, whereas the C=N double bond is still (*E*)-configured. Recrystallization from *t-*butanol provided the pure (*E,E*)-configured betalamic acid derivative (*E,E*)-**23** (Figure 1).

In Figure 2, the second strategy for the synthesis of betalamic acid (**9**), developed by Bϋchi et al. [36], is displayed. This approach started from benzylnorteleoidine **VI** obtained by Robinson–Schöpf condensation. The first reaction includes the protection of the diol by formation of a cyclic *ortho* ester. Hydrogenolytic cleavage of the N-benzyl protective group provided the secondary amine **VII**. Reaction of the aminoketone **VII** with allyl magnesium chloride yielded the tertiary alcohol **VIII** with high diastereoselectivity. The secondary amine **VIII** was then converted into *O*-benzoylhydroxylamine **IX.** First, amine **IX** was neutralized with K_2_CO_3_ and reacted with dibenzoyl peroxide in DMF, leading to formation of the protected amine. Subsequently, acetylation of the alcohol provided *O*-benzoylhydroxylamine **IX**. Next, the *ortho* ester was cleaved with oxalic acid in water to obtain diol **X**. This latter compound was then oxidized with *N*-chlorosuccinimide (NCS) and dimethylsulfide to achieve the diketone **XI**. Ozonolysis of **XI** led to the formation of aldehyde **XII**. Treatment of **XII** with lead tetraacetate in benzene and methanol converted the diketone moiety into an unstable dicarboxylic acid, which, upon loss of HOAc and BzOH, yielded (±)-betalamic acid (**9**) as a mixture of (*E*)- and (*Z*)-configured diastereomers after silica gel chromatography [36].

Despite the fact that two methods for the synthesis of betalamic acid (**9**) and its derivatives have been reported in the literature, the majority of betalamic acid (**9**) is produced through extraction from pigments, followed by basic hydrolysis.

To investigate relationships between the chemical structure and biological properties of indicaxanthin derivatives in further detail, analogs **13** of **12** that lack the two carboxy moieties in positions C-2- and C-6 were considered first. Herein, we describe the design and synthesis of betalamic acid analog **13** that is devoid of carboxy groups in positions C-2 and C-6. Additionally, experiments were conducted to synthesize the betalamic acid derivative **14** in a simpler and more cost-effective manner and to evaluate its reactivity toward oxygen (Figure 3).

## 2. Results and Discussion

### 2.1. Chemistry

The plan for the synthesis of **13**, the analog of betalamic acid without carboxy groups in positions C-2- and C-6, is outlined in Figure 4.

We planned to synthesize **13** from the α,β-unsaturated ester **15** that bears a Boc-protective group at the piperidine ring. At first, the ester must be reduced to afford an aldehyde and finally, the Boc-protective group must be removed. The α,β-unsaturated ester **15** can be obtained by a *Wittig* reaction of α-bromoketone **17** and the subsequent β-elimination of **16**. The α-bromoketone should be available by α-bromination of an appropriate piperidone derivative, e.g., **18**.

The synthesis started with piperidine **19** (Figure 5), which was protected with (Boc)_2_O to afford Boc-protected piperidone **18**. In order to introduce a double bond in positions C-5 and C-6 of the piperidine ring, piperidone **18** was brominated in the α-position using Br_2_ and AlCl_3_ to generate the α-bromoketone **17** in a 46% yield [38]. The conjugated double bond system is a characteristic feature of the class of betalains. Thus, the first double bond was introduced by a Wittig reaction of the α-bromoketone **17** with Ph_3_P=CHCO_2_Et to give the α,β-unsaturated ester **16** in a 95% yield [39]. Although the formation of (*E*)/(*Z*)-diastereomers was expected, the ^1^H and ^13^C NMR spectra reveal only one set of signals, indicating a single diastereomer, presumably (*E)*-**16**. LiBr and Li_2_CO_3_ were used to induce dehydrobromination (β-elimination), resulting in the formation of completely conjugated compound **15**, which was isolated in a 88% yield [40]. The ^1^H NMR spectrum of **15** reveals two distinct sets of signals, indicating the presence of (*E*)- and (*Z*)-configured esters **15** in the ratio 9:1. Since diastereomeric (*E*)- and (*Z*)-configured esters **15** could not be separated by flash column chromatography, the mixture was used to prepare the aldehyde **21**. According to the first theory, aldehyde **21** should be obtained directly by the reduction of the ester **15** with DIBAL-H. However, even at −78 °C in toluene, only the primary alcohol **20** was formed and isolated in a 94% yield. Alternatively, the primary alcohol **20** was synthesized by the reduction of the ester **15** with LiAlH_4_. Several methods have been reported in the literature for the oxidation of primary alcohols to aldehydes [41]. A method with broad applicability and high yields is the Dess–Martin periodinane (DMP) oxidation method. Unexpectedly, the oxidation of allyl alcohol **20** with DMP resulted in low yields of the product, which was difficult to purify. Therefore, the alcohol **20** was oxidized via radical oxidation with TEMPO [41] and CuCl to provide the aldehyde **21** in a 76% yield. To obtain the aldehyde **13** as an analog of betalamic acid (**9**), the Boc-protective group of **21** was removed. Unfortunately, removing the Boc-protective group under typical conditions with F_3_CCO_2_H did not result in the desired aldehyde **13**. Several methods were investigated to remove the Boc-protective group from **21** to achieve **13**. In the end, a rather unusual method, i.e., heating the Boc-protected compound **21** in a mixture of water and dioxane under neutral conditions [42], was successful. Due to the instability of the secondary amine **13**, the isolated yield of **13** was rather low. In particular, condensation and polymerization reactions, as well as oxidation processes, were observed during the purification process. Despite the instability, ^1^H and ^13^C NMR spectra could be recorded to identify and characterize **13**.

In addition to betalamic acid analog **13,** 1,2,3,4-tetrahydropyridine derivatives **22** and **23** were designed and synthesized (Figure 6). The reactivity of these 1,2,3,4-tetrahydropyridines **22** and **23** and further analogs towards oxygen should be investigated. The key intermediate for the synthesis of **22** and **23** is 4-methylenepiperidine **24**, which can be obtained by double allylation of iminodiacetic acid diester **25** with dichloride **26,** as reported by Einhorn et al. [43]. Transformation of the methylene moiety of **24** into a ketone and subsequent introduction of a double bond in the ring result in the formation of **23**. The α,β-unsaturated ester **22** can prepared by an additional *Wittig* reaction of a ketone intermediate.

For the synthesis of methylenepiperidine **24**, the diester **25** and the diiodide **29** were prepared (Figure 7). The diester HCl salt **28** was obtained by esterification of iminodiacetic acid (**27)** with SOCl_2_ in refluxing ethanol. The secondary amine of **27** was protected with Boc_2_O to afford the carbamate **25** in a 76% yield. The diester **25** was initially treated with dichloride **26**, which, however, did not lead to the desired 4-methylenepiperidine **24**. To obtain the desired methylenepiperidine **24**, the more reactive diiodide **29** should be employed instead of the dichloride **26**. Allyl diiodide **29** was freshly prepared by a *Finkelstein* reaction of commercially available 3-chloro-2-(chloromethyl)prop-1-ene (**26**) with NaI in acetone [44]. After a reaction time of 16 h in refluxing acetone, the diiodide **29** was obtained in a 99% yield.

For the double allylation of diester **25**, LDA was generated in situ from *n*-BuLi and *i*-Pr_2_NH. Deprotonation of diester **25** with freshly prepared LDA and subsequent treatment with diiodide **29** provided the methylenepiperidine **24** in a 77% yield. The IR and ^1^H NMR spectra of piperidine **24** demonstrate the successful synthesis of the piperidine ring. A band at 1655 cm^−1^ in the IR spectrum originates from the C=C stretching vibration. Two sets of signals can be found in the ^1^H NMR spectrum, as illustrated by two singlets for the protons of the exocyclic methylene moiety (R_2_C=CH_2_) at 4.83 and 4.92 ppm and two singlets for the Boc group at 1.42 and 1.47 ppm. These signal pairs confirm the formation of *trans*- and *cis*-configured diastereomers *trans*-**24** and *cis*-**24**, which are present in the ratio 9:1. *Lemieux*–*Johnson* oxidation using catalytic amounts of OsO_4_ and an excess of NaIO_4_ transformed the 4-methylenepiperidine **24** into piperidinone **30** [45]. Despite the fact that compound **24** was used as a mixture of diastereomers, only one diastereomer could be observed for compound **30**. The subsequent *Wittig* reaction of ketone **30** provided the α,β-unsaturated ester **31**, which shows an even higher structural similarity to betalamic acid than methylenepiperidine **24** and piperidinone **30** (Figure 8).

Since the piperidines **24**, **30**, and **31** do not contain a halogen atom for elimination, another method for the introduction of a double bond into the piperidine ring was required. For this purpose, the Boc-protective group was removed, yielding the secondary amines **32**, **33** and **34**. The secondary amines **32–34** were reacted with in situ prepared *t*-BuOCl followed by base-induced HCl elimination, according to the method reported by Zhong et al. [46] (Figure 9). For compound **32**, isolation of the expected product **35** was not possible due to the fast oxidation to its pyridine form **38,** isolated in a 6% yield. For compound **33,** the formation of the conjugate system was successful, leading to the desired product **36** in a 39% yield. For this product, we did not observe the formation of the pyridine form **39**. With compound **34,** the conjugate derivative **37** was obtained. Although, a slow conversion to the pyridine form **40** was observed.

### 2.2. Antioxidant Activity and Stability

Due to the instability of aldehyde **13**, we decided to evaluate the total antioxidant activity (TAC) of the protected aldehyde **21**. For this purpose, the Folin–Ciocalteu assay was employed [47]. This method can be classified among the protocols used to evaluate the TAC in the electron transfer (ET) group [48]. Reduction of the oxidant leads to a change in its properties, such as light absorption or fluorescence, which are measured using spectroscopy techniques [49]. In the Folin–Ciocalteu assay, a molybdotungstophosphate heteropolyanion (3H_2_O-P_2_O_5_-14WO_3_-4MoO_3_-10H_2_O) is used for the oxidation of phenolic compounds in basic solution (carbonate buffer). The reduction leads to a colored product with an absorption maximum (λ_max_) at 765 nm. The molibdenum center in the complex is reduced from Mo(VI) to Mo(V) by an e^−^ donated from the antioxidant, leading to a blue solution [49].

Unfortunately, during the test of the protected aldehyde **21** in the Folin–Ciocalteu assay, a change in the color of the solution could not be recorded as reduction of the molybdenum complex did not take place. All information were provided in Appendix A.

The stability towards oxygen of the 1,2,3,4-tetrahydropyridines **35–37** was observed spectroscopically using ^1^H NMR spectra. After oxidizing the 4-methylenepiperidine **32** with *t*-BuOCl, only pyridine **38** was detected, indicating the fast oxidation of intermediate tetrahydropyridine **35** by O_2_. In contrast to the fast oxidation of methylenetetrahydropyridine **35**, tetrahydropyridone **36** did not show any potential to be further oxidized. The most promising properties were observed for the ester **37**. Although it could be isolated in its pure form, recording of ^1^H NMR spectra over a period of several days revealed the slow oxidation of ester **37** to pyridine **40** (Figure 3).

## 3. Materials and Methods

### 3.1. Chemistry

Moisture and oxygen sensitive reactions were carried out under nitrogen, dried with molecular sieves (3 or 4 Å, 8 to 12 mesh, Acros Organics), in dry glassware (Schlenk flasks or Schlenk tubes, sealed with rubber septa). All solvents were of analytical grade quality. Flash chromatography (FC): silica gel 60, 40–63 µm (Machery Nagel); parentheses include: diameter of the column (*Ø*), length of the stationary phase (h), fraction size (*V*) and eluent. Melting point: melting point system MP50 (Mettler Toledo, gießen, Germany), open capillary, uncorrected. MS: MicroTOFQII mass spectrometer (Bruker Daltonics, Bremen, Germany); deviations of the found exact masses from the calculated exact masses were 5 ppm or less; the data were analyzed with DataAnalysis^®^ (Bruker Daltonics). NMR: NMR spectra were recorded in deuterated solvents on Agilent DD2 400 MHz and 600 MHz NMR spectrometers (Agilent, Santa Clara, CA, USA); chemical shifts (*δ*) are reported in parts per million (ppm) against the reference substance tetramethylsilane and calculated using the solvent residual peak of the undeuterated solvent; coupling constants are given with 0.5 Hz resolution; assignment of ^1^H and ^13^C NMR signals was supported by 2D NMR techniques where necessary. IR: FT/IR IR Affinity^®^-1 spectrometer (Shimadzu, Düsseldorf, Germany) using ATR technique. Characterization data including ^1^H and ^13^C NMR spectra for synthesized compounds are reported in Appendix A.

#### 3.1.1. Synthesis of *Tert*-Butyl 4-Oxopiperidine-1-Carboxylate (**18**)

Piperidin-4-one monohydrate hydrochloride **19** (5.0 g, 32.5 mmol, 1.0 eq.) was dissolved in a 1:1 mixture of THF:H_2_O (100 mL) at room temperature. NaHCO_3_ (5.47 g, 65 mmol, 2.0 eq.) was added and the mixture was stirred for 15 min at rt. Afterwards, Boc_2_O (8.52 g, 39 mmol, 1.2 eq.) was added and the mixture was stirred for 16 h at room temperature. The mixture was diluted with Et_2_O (50 mL) and washed with aqueous solution of KHSO_4_ 5% (3 × 50 mL), H_2_O (3 × 50 mL) and brine (3 × 50 mL). The combined organic layers were concentrated in vacuo. The residue was purified by flash column chromatography (petroleum ether/EtOAc 9/1 → 5/5, *R*_f_ = 0.34 (cHex/EtOAc 7:3)). Colorless solid, mp 73–76 °C, yield 6.22 g (31 mmol, 96%). Exact mass (APCI): *m/z* = 200.1279 (calcd.200.1287 for C_10_H_18_NO_3_^+^ [M+H^+^]).^1^H NMR (600 MHz, DMSO-*d*_6_): *δ* (ppm) = 1.42–1.44 (m, 9H, C(C*H*_3_)_3_), 2.34 (t, *J* = 6.2 Hz, 4H, 3-(CH_2_)_2_), 3.60 (t, *J* = 6.2 Hz, 4H, 2-(CH_2_)_2_). ^13^C NMR (151 MHz, DMSO-*d*_6_): *δ* (ppm) = 28.0 (3C, C(*C*H_3_)_3_), 40.0 (2C, C-3), 40.3 (2C, C-2), 79.2 (1C, O*C*(CH_3_)_3_), 153.8 (1C, *C(=O)O*C(CH_3_)_3_), 207.4 (1C, R_2_*C*=O). FT-IR (neat): *ṽ* (cm^−1^) = 2985, 2870 (C-H, aliph.), 1724 (C=O_carb._), 1678 (C=O_ketone_), 1161 (C-O).

#### 3.1.2. Synthesis of *Tert*-Butyl 3-Bromo-4-Oxopiperidine-1-Carboxylate (**17**)

1-Boc-piperidin-4-one **18** (10 g, 50 mmol, 1.0 eq.) was dissolved in THF (30 mL) and Et_2_O (30 mL). AlCl_3_ (0.67 g, 5.0 mmol, 0.1 eq.) was added and at 0 °C, Br_2_ (2.6 mL, 50 mmol, 1.0 eq.) was added slowly over a period of 30 min. Afterwards, the solution was stirred at 0 °C for 18 h. Afterwards, the formed solid was filtered off and washed with Et_2_O. The organic layer was dried (Na_2_SO_4_), filtered and concentrated in vacuo. The residue was purified by flash column chromatography (petroleum ether/EtOAc 9/1 → 5/5, *R*_f_ = 0.41 (cHex/EtOAc 7:3)). Colorless solid, mp 90–93 °C, yield 6.42 g (23 mmol, 46%). Exact mass (APCI): *m/z* = 278.0329 (calcd.278.0392 for C_10_H_17_BrNO_3_^+^ [M+H^+^]). ^1^H NMR (200 MHz, DMSO-d6): *δ* (ppm) = 1.44 (s, 9H, C(CH_3_)_3_), 2.50–2.52 (m, 1H, 5-H), 2.70–2.78 (m, 1H, 5-H), 3.58–3.68 (m, 3H, 2 × 6-H, 2-H), 3.98–4.08 (m, 1H, 2-H), 4.77 (s, 1H, 3-H). 13C NMR (50 MHz, DMSO-d6): δ (ppm) = 27.9 (3C, C(CH_3_)_3_), 35.8 (1C, C-5), 42.7 (1C, C-6), 47.7 (1C, C-2), 49.0 (1C, C-3), 79.8 (1C, OC(CH3)3), 153.8 (1C, C(=O)-OC(CH3)3), 199.7 (1C, R2C=Oketone). FT-IR (neat): *ṽ* (cm^−1^) = 2978, 2931 (C-H. aliph.), 1724 (C=O_ketone_), 1674 (C=O_carbamate_), 1157 (C-O), 648 (C-Br).

#### 3.1.3. Synthesis of *Tert*-Butyl (*E*)-3-Bromo-4-(Ethoxycarbonylmethylene)Piperidine-1-CarBoxylate (**16**)

4-(Ethoxycarbonylmethylene)triphenylphosphorane (6.9 g, 20 mmol, 1.1 eq.) was added to a solution of *tert*-butyl 3-bromo-4-oxopiperidine-1-carboxylate **17** (5.03 g, 18 mmol, 1.0 eq.) in CH_2_Cl_2_ (450 mL) and the reaction mixture was stirred at reflux for 2 h. Then, the mixture was cooled to room temperature and concentrated in vacuo. The residue was purified by flash column chromatography (petroleum ether/EtOAc 9/1 → 7/3, *R*_f_ = 0.57 (cHex/EtOAc 7:3)). Colorless solid, mp 114–115 °C, yield 5.98 g (17 mmol, 95%). Exact mass (APCI): *m/z* = 348.0777 (calcd.348.0810 for C_14_H_23_BrNO_4_^+^ [M+H^+^]). ^1^H NMR (200 MHz, DMSO-*d*_6_): *δ* (ppm) = 1.21 (t, *J* = 7.1 Hz, 3H, OCH_2_C*H*_3_), 1.42 (s, 9H, C(CH_3_)_3_), 2.56–2.88 (m, 2H, 5-CH_2_, 6-CH_2_), 3.34–3.51 (m, 2H, 2-CH_2_, 5-CH_2_), 4.01–4.29 (m, 4H, OC*H*_2_CH_3_, 6-CH_2_, 2-CH_2_), 5.04–5.12 (m, 1H, 3-CH), 6.12 (s, 1H, R_2_C=C*H*). ^13^C NMR (50 MHz, DMSO-*d*_6_): *δ* (ppm) = 14.0 (1C, OCH_2_*C*H_3_), 24.1 (1C, C-5), 27.9 (3C, C(*C*H_3_)_3_), 42.5 (1C, C-6), 51.0 (1C, C-2), 53.3 (1C, C-3), 59.9 (1C, O*C*H_2_CH_3_), 79.2 (1C, *C*(CH_3_)_3_), 113.8 (1C, R_2_C=*C*H), 153.7 (1C, *C(=O)*OC(CH_3_)_3_), 154.0 (1C, C-4), 165.1 (1C, *C*O_2_Et). Only one set of signals can be observed in the spectra. FT-IR (neat): *ṽ* (cm^−1^) = 2985, 2920 (C-H, aliph.), 1712 (C=O), 1670 (C=O), 1654 (C=C), 1161 (C-O), 641 (C-Br).

#### 3.1.4. Synthesis of *Tert*-Butyl (*E*)- and (*Z*)-Ethoxycarbonylmethylene)-3,4-Dihydropyridine-1(2H)-Carboxylate (**15**)

Ester **16** (5.74 g, 16 mmol, 1.0 eq) was dissolved in dry DMF (165 mL). LiBr (8.6 g, 99 mmol, 6.0 eq.) and Li_2_CO_3_ (7.31 g, 99 mmol, 6.0 eq.) were added and the solution was stirred at 75 °C for 3 h. Then, the mixture was cooled to room temperature and extracted with EtOAc (3 × 100 mL). The combined organic layers were concentrated in vacuo. The residue was purified by flash column chromatography (petroleum ether/EtOAc 9/1 → 7/3, *R*_f_ = 0.72 (cHex/EtOAc 7:3)). Yellow oil, yield 3.88 g (14 mmol, 88%). Exact mass (APCI): *m/z* = 268.1497 (calcd. 268.1549 for C_14_H_22_NO_4_^+^ [M+H^+^]). Compound **15** was isolated as a mixture of ((*E*):(*Z*)) isomers. In the NMR spectra, a ratio of 9:1 is observed. ^1^H NMR (600 MHz, DMSO-*d*_6_): *δ* (ppm) = 1.19 (t, *J* = 7.1 Hz, 3H, OCH_2_C*H*_3_), 1.46 (s, 9H, C(CH_3_)_3_), 2.49–2.53 (m, 0.2H, 5-CH_2_), 3.00–3.07 (m, 1.8H, 5-CH_2_), 3.52–3.60 (m, 2H, 6-CH_2_), 4.06 (q, *J* = 7.1 Hz, 2H, OC*H*_2_CH_3_), 5.32–5.34 (m, 0.1H, R_2_C=C*H*), 5.47–5.53 (m, 0.9H, 3-CH) 5.55–5.59 (m, 0.9H, R_2_C=C*H*), 6.54–6.61 (m, 0.1H, 3-CH) 7.00–7.13 (m, 0.9H, 2-CH), 7.14–7.16 (m, 0.1H, 2-CH). ^13^C NMR (151 MHz, DMSO-*d*_6_): *δ* (ppm) = 14.2 (1C, OCH_2_*C*H_3_), 24.8 (0.9C, C*-5*), 27.7 (3C, C(*C*H_3_)_3_), 30.0 (0.1C, C-5), 40.0 (1C, C-6), 59.0 (1C, O*C*H_2_CH_3_), 81.5 (1C, *C*(CH_3_)_3_), 103.4 (0.1C, C-3), 108.1 (0.9C, C-3), 109.6 (0.1C, R_2_C=*C*H) 110.6 (0.9C, R_2_C=*C*H), 132.5 (0.9C, C-2), 133.0 (0.1C, C-2), 147.5 (1C, *C(=O)*OC(CH_3_)_3_), 147.9 (0.1C, R_2_*C*=CH) 149.1 (0.9C, R_2_*C*=CH), 166.0 (1C, *C*O_2_Et) FT-IR (neat): *ṽ* (cm^−1^) = 2978, 2931, 2900 (C-H, aliph.), 1708 (C=O), 1700 (C=O), 1608 (C=C), 1145 (C-O), 1111 (C-O).

#### 3.1.5. Synthesis of *Tert*-Butyl (*E*)- and (*Z*)-4-(2-Hydroxyethylidene)-3,4-Dihydropyridine-1(2H)-Carboxylate (**20**)

##### Procedure 1

Under N_2_, **15** (2.80 g, 10.5 mmol, 1.0 eq.) was dissolved in dry toluene (25 mL). The solution was cooled to −78 °C, DIBAL-H (1 m solution in hexane, 31.5 mL, 31.5 mmol, 3.0 eq.) was added dropwise within 15 min, and the reaction mixture was stirred at −78 °C for 40 min. Then, at −78 °C, CH_3_OH (2 mL) was added carefully, and the mixture was warmed to room temperature. The mixture was filtered, the solid was washed with EtOAc (3 × 30 mL) and the combined organic layers were concentrated in vacuo. Yellow oil, yield 2.22 g (9.9 mmol, 94%).

##### Procedure 2

Under N_2_, **15** (570 mg, 2.1 mmol, 1.0 eq.) was dissolved in dry THF (5 mL) and at −10 °C, LiAlH_4_ (160 mg, 4.3 mmol, 2.0 eq.) was added and the mixture was stirred for 40 min. A saturated solution of potassium sodium tartrate (5 mL) was added, and the mixture was extracted with EtOAc (3 × 15 mL). The combined organic layers were dried (Na_2_SO_4_), filtered and concentrated in vacuo. The residue was purified by flash column chromatography (cHex/EtOAc 9/1 → 6/4, *R*_f_ = 0.45 (petroleum ether/EtOAc 7:3)). Yellow oil, yield 0.33 g (1.5 mmol, 70%). Exact mass (APCI): *m/z* = 226.1383 (calcd. 226.1443 for C_12_H_20_NO_3_^+^ [M+H^+^]). Compound **20** was isolated as a mixture of ((*E*):(*Z*)) isomers. In the NMR spectra, a ratio of 8.5:1.5 is observed. Peaks of minor isomer are not all clearly visible in the spectra. ^1^H NMR (600 MHz, DMSO-*d*_6_): *δ* (ppm) = 1.44 (s, 9H, C(CH_3_)_3_), 2.36–2.44 (m, 2H, 5-CH_2_), 3.48–3.55 (m, 2H, 6*-*CH*_2_*), 3.96–4.01 (m, 2H, C*H*_2_OH), 4.53–4.58 (m, 1H, O*H*), 5.13 (t, J = 6.8 Hz, 0.15H, R_2_C=C*H*), 5.35 (t, *J* = 6.8 Hz, 0.85H, R_2_C=C*H*), 5.40–5.46 (m, 1H, 3-CH), 6.65–6.78 (m, 1H, 2-CH). ^13^C NMR (151 MHz, DMSO-*d*_6_): *δ* (ppm) = 26.9 (1C, *C*-5), 30.9 (3C, C(*C*H_3_)_3_), 43.1 (1C, C-6), 59.9 (1C, *C*H_2_OH), 83.6 (1C, *C*(CH_3_)_3_), 112.7 (1C, C-3), 127.8 (1C, R_2_C=*C*H), 128.4 (1C, C-2), 133.2 (1C, R_2_*C*=CH), 154.3 (1C, *C(=O)*OC(CH_3_)_3_). FT-IR (neat): *ṽ* (cm^−1^) = 3398 (O-H), 2974, 2931, 2873 (C-H, aliph.), 1701 (C=O), 1643 (C=C), 1612 (C=C), 1161, 1141 (C-O).

#### 3.1.6. Synthesis of *Tert*-Butyl (*E*)- and (*Z*)-4-(Formylmethylene)-3,4-Dihydropyridine-1(2H)-Carboxylate (**21**)

CuCl (11 mg, 0.12 mmol, 0.1 eq.) and TEMPO (17 mg, 0.12 mmol, 0.1 eq.) were added to a solution of racemic mixture of allylic alcohol **20** (270 mg, 1.20 mmol, 1.0 eq.) in dry DMF (3 mL). The solution was stirred at room temperature for 16 h. Afterwards, the solution was poured into water/ice slowly and the mixture was warmed to room temperature. The solid was filtered, washed with H_2_O and dried. Purification by flash column chromatography (petroleum ether/EtOAc 9/1 → 7/3 *R*_f__a_ = 0.38, *R*_f__b_ = 0.30 (cHex/EtOAc 7:3)). Yellow oil, yield 240 mg (1.07 mmol, 90%). Exact mass (APCI): *m/z* = 224.1208 (calcd. 224.2495 for C_24_H_35_N_2_O_6_^+^ [M+H^+^]). Compound **21** was isolated as a mixture of ((*E*):(*Z*)) isomers. In the NMR spectra, a ratio of 6:4 is observed. Peaks of minor isomer are not all clearly visible in the spectra. ^1^H NMR (600 MHz, CDCl_3_): *δ* (ppm) = 1.52 (s, 3.6H, C(C*H*_3_)_3_), 1.53 (s, 5.4H, C(C*H*_3_)_3_), 2.64 (t, *J* = 6.8 Hz, 0.8H, 5-CH_2_), 2.91–3.14 (m, 1.2H, 5-CH_2_), 3.74 (t, *J* = 6.8 Hz, 2H, 6-CH_2_), 5.44–5.53 (m, 0.6H, 3-CH), 5.57 (d, *J* = 7.8 Hz, 0.4H, R_2_C=C*H*), 5.78 (d, *J* = 7.7 Hz, 0.6H, R_2_C=C*H*), 6.23–6.36 (m, 0.4H, 3-CH), 7.13–7.19 (m, 0.6H, 2-CH), 7.27–7.41 (m, 0.4H, 2-CH), 9.94 (d, *J* = 7.9 Hz, 0.6H, C*H*O), 10.05 (d, *J* = 7.9 Hz, 0.4H, C*H*O). ^13^C NMR (151 MHz, CDCl_3_): *δ* (ppm) = 24.7 (0.6C, C-5), 28.2 (3C, C(*C*H_3_)_3_), 30.9 (0.4C, C-5), 40.4 (1C, C-6), 82.8 (1C, *C*(CH_3_)_3_), 101.2 (0.4C, C-3), 108.1 (0.6C, C-3), 121.5 (1C, R_2_C=*C*H), 134.0 (0.6C, C-2), 134.5 (0.4C, C-2), 151.1 (1C, R_2_*C*=CH), 151.3 (1C, *C*(=O)OC(CH_3_)_3_), 189.6 (0.4C, CHO), 190.0 (0.6C, CHO). FT-IR (neat): *ṽ* (cm^−1^) = 3062, 2966, 2877 (C-H, aliph.), 1712 (C=O), 1647 (C=C), 1593 (C=C), 1141, 1122 (C-O).

#### 3.1.7. Synthesis of (*E*)- and (*Z*)-2-[2,3-Dihydropyridin-4(1H)-Ylidene]Acetaldehyde (**13**)

A solution of **21** (110 mg, 0.49 mmol, 1.0 eq.) in water (9 mL) and 1,4-dioxane (1 mL) was heated to 85 °C for 2 h. Then, the solution was cooled to room temperature and the mixture was extracted with EtOAc (6 × 15 mL). The combined organic layers were dried (Na_2_SO_4_), filtered and concentrated in vacuo. The residue was purified by flash column chromatography (petroleum ether/EtOAc 9/1 → 1/9 *R*_f__a_ = 0.15 (EtOAc)). Yellow/orange oil, yield 12 mg (0.10 mmol, 20%). The compound is highly unstable and quickly decomposed during purification. Aldehyde **13** was isolated as a mixture of ((*E*):(*Z*)) isomers. In the NMR spectra, a ratio of 6:4 is observed. Peaks of minor isomer are not all clearly visible in the spectra. ^1^H NMR (200 MHz, CDCl_3_): *δ* (ppm) = 2.61 (t, *J* = 7.0 Hz, 0.8H, 5-CH_2_), 3.00–3.12 (m, 1.2H, 5-CH_2_), 3.31–3.46 (m, 2H, 6-CH_2_), 4.7–4.81 (m, 1H, NH), 5.19 (d, *J* = 7.2 Hz, 0.6H, R_2_C=C*H*), 5.29 (d, *J* = 7.9 Hz, 0.4H, R_2_C=C*H*), 5.60 (d, *J* = 8.3 Hz, 0.6H, 3-C*H*), 6.00 (d, *J* = 7.5 Hz, 0.4H, 3-CH), 6.66–6.77 (m, 1H, 2-CH), 9.79 (d, *J* = 8.4 Hz, 0.6H, CHO), 9.93 (d, *J* = 8.0 Hz, 0.4H, CHO). ^13^C NMR (50 MHz, CDCl_3_): *δ* (ppm) = 25.3 (0.6C, C*-*5), 31.75 (0.4C, C-5), 40.5 (0.6C, C-6), 40.9 (0.4C, C-6), 93.9 (0.4C, C-3), 100.3 (0.6C, C-3), 116.2 (0.4C, R_2_C=*C*H), 117.1 (0.6C, R_2_C=*C*H), 142.6 (1C, C-2), 143.4 (1C, C-2), 155.3 (0.4C, R_2_*C*=CH), 171.3 (0.6C, R_2_*C*=CH), 189.2 (0.4C, *C*HO), 189.3 (0.6C, *C*HO).

#### 3.1.8. Synthesis of Diethyl 2,2′-Iminodiacetate∙HCl (**28**)

At 0 °C, SOCl_2_ (20.0 mL, 275 mmol, 1.5 eq) was added dropwise to a suspension of iminodiacetic acid **27** (24.4 g, 184 mmol, 1.0 eq) in EtOH abs. (200 mL). Afterwards, the reaction mixture was heated to reflux for 16 h. The solution was cooled down to room temperature and concentrated in vacuo. Colorless solid, mp 88–89 °C, yield 40.6 g (98%). C_8_H_16_ClNO_4_ (225.7 g/mol). ^1^H NMR (600 MHz, DMSO-*d*_6_): *δ* (ppm) = 1.24 (t, *J* = 7.1 Hz, 6H, 2 × OCH_2_C*H*_3_), 3.70 (brs, 2H, NH_2_^+^), 3.99 (s, 4H, 2 × C*H*_2_), 4.21 (q, *J* = 7.1 Hz, 4H, 2 × OC*H*_2_CH_3_). ^13^C NMR (151 MHz, DMSO-*d*_6_): *δ* (ppm) = 13.9 (2C, 2 × OCH_2_*C*H_3_), 46.4 (2C, 2 × *C*H_2_), 61.8 (2C, 2 × O*C*H_2_CH_3_), 166.4 (2C, 2 × O=*C*OEt). IR (neat): *ṽ* (cm^−1^) = 2936 (C-H_alip_), 1736 (C=O_ester_), 1204, 1076, 1015 (C-N, C-O). Exact mass (APCI): *m/z* = 190.1073 (calcd. 190.1074 for C_8_H_16_NO_4_ [M-Cl]^+^).

#### 3.1.9. Synthesis of Diethyl 2,2′-[N-(*Tert*-Butoxycarboxyl)Imino]Diacetate (**25**)

NaHCO_3_ (22.8 g, 271 mmol, 3.0 eq) was added to a solution of **28** (20.4 g, 90.4 mmol, 1.0 eq) in THF (80 mL) and H_2_O (20 mL). The reaction mixture was stirred for 30 min at room temperature. After the addition of Boc_2_O (19.3 mL, 90.4 mmol, 1.0 eq), the mixture was stirred at room temperature for 16 h. Then, it was extracted with EtOAc (3 × 50 mL) and the combined organic layers were dried (Na_2_SO_4_), filtered and concentrated in vacuo. The residue was purified by flash column chromatography (ø = 8 cm, *h* = 16 cm, cHex/EtOAc 8:2, *V* = 80 mL). Colorless oil, yield 19.9 g (76%). C_13_H_23_NO_6_ (289.3 g/mol). TLC: *R*_f_ = 0.36 (cHex/EtOAc 8:2). ^1^H NMR (400 MHz, DMSO-*d*_6_): *δ* (ppm) = 1.19 (t, *J* = 7.1 Hz, 3H, OCH_2_C*H*_3_), 1.20 (t, *J* = 7.1 Hz, 3H, OCH_2_C*H*_3_), 1.35 (s, 9H, C(C*H*_3_)_3_), 3.98 (s, 2H, NC*H*_2_), 4.01 (s, 2H, NC*H*_2_), 4.10 (q, *J* = 7.1 Hz, 2H, OC*H*_2_CH_3_), 4.12 (q, *J* = 7.1 Hz, 2H, OC*H*_2_CH_3_). Ratio of rotamers is 1:1. ^13^C NMR (151 MHz, DMSO-*d*_6_): *δ* (ppm) = 14.0 (1C, OCH_2_*C*H_3_), 14.1 (1C, OCH_2_*C*H_3_), 27.7 (3C, C(*C*H_3_)_3_), 49.2 (1C, *C*H_2_), 49.7 (1C, *C*H_2_), 60.4 (2C, O*C*H_2_CH_3_), 79.9 (1C, *C*(CH_3_)_3_), 154.6 (1C, N(*C*=O)O), 169.45 (1C, O=*C*OEt), 169.53 (1C, O=*C*OEt). Ratio of rotamers is 1:1. IR (neat): *ṽ* (cm^−1^) = 2978 (CH_aliph_), 1747 (C=O_ester_), 1700 (C=O_carbamate_), 1185, 1159, 1026 (C-N, C-O). Exact mass (APCI): *m/z* = 290.1587 (calcd. 290.1598 for C_13_H_24_NO_6_^+^ [M+H]^+^).

#### 3.1.10. Synthesis of 3-Iodo-2-(Iodomethyl)Prop-1-Ene (**29**) [44]

NaI (17.8 g, 119 mmol, 2.5 eq) was added to a solution of 3-chloro-2-(chloromethyl)prop-1-ene **26** (5.50 mL, 47.5 mmol, 1.0 eq) in acetone (100 mL) and the mixture was stirred at reflux for 16 h. The suspension was cooled to room temperature and concentrated in vacuo. The residue was dissolved in H_2_O (75 mL) and cHex (75 mL). After separation of the two layers, the organic layer was washed with Na_2_SO_3_ (2 × 50 mL) and H_2_O (50 mL), dried (Na_2_SO_4_), filtered and concentrated in vacuo. Light green solid, mp 28–29 °C, yield 14.5 g (99%). C_4_H_6_I_2_ (307.9 g/mol). ^1^H NMR (600 MHz, CD_3_OD): *δ* (ppm) = 4.21 (s, 4H, 2 × C*H*_2_I), 5.40 (s, 2H, R_2_C=C*H*_2_). ^13^C NMR (151 MHz, CD_3_OD): *δ* (ppm) = 6.6 (2C, 2 × *C*H_2_I), 116.4 (1C, R_2_C=*C*H_2_), 146.0 (1C, R_2_*C*=CH_2_). Exact mass (APCI): *m/z* = 308.8637 (calcd. 308.8632 for C_4_H_7_I_2_^+^ [M+H]^+^).

#### 3.1.11. Synthesis of 1-*Tert*-Butyl 2,6-Diethyl *Cis*- and *Trans*-4-Methylenepiperidine-1,2,6-Tricarboxylate (**24**)

At −78 °C, *n*-BuLi (1.6 m in *n*-hexane, 74.4 mL, 119 mmol, 2.1 eq) was added dropwise to a solution of *i-*Pr_2_NH (16.7 mL, 119 mmol, 2.1 eq) in dry THF (170 mL). After the mixture was stirred for 1 h, a solution of **25** (16.4 g, 56.7 mmol, 1.0 eq) in dry THF (15 mL) was added and the mixture was stirred for 1 h at −78 °C. Then, a solution of **29** (22.1 g, 68.1 mmol, 1.2 eq) in dry THF (15 mL) was added and the reaction mixture was stirred for 30 min at −78 °C and warmed up to room temperature over 16 h. At 0 °C, H_2_O (150 mL) was added and the mixture was extracted with EtOAc (3 × 80 mL). The combined organic layers were dried (Na_2_SO_4_), filtered and concentrated in vacuo. The residue was purified twice by flash column chromatography (1. ø = 8 cm, *h* = 25 cm, cHex/EtOAc 9:1 → 8:2, *V* = 80 mL, 2. ø = 8 cm, *h* = 25 cm, cHex/EtOAc 9:1 → 8:2, *V* = 80 mL). Yellow oil, yield 14.8 g (77%). C_17_H_27_NO_6_ (341.4 g/mol). TLC: *R*_f_ = 0.26 (cHex/EtOAc 8:2). ^1^H NMR (600 MHz, CDCl_3_): *δ* (ppm) = 1.25 (t, *J* = 7.1 Hz, 3H, OCH_2_C*H*_3_), 1.26 (t, *J* = 7.1 Hz, 3H, OCH_2_C*H*_3_), 1.42 (s, 9 × 0.9H, C(C*H*_3_)_3_), 1.47 (s, 9 × 0.1H, C(C*H*_3_)_3_*), 2.43–2.51 (m, 2 × 0.1H, 3/5-C*H*_2_*), 2.65 (dd, *J* = 15.9/2.7 Hz, 1H, 3/5-C*H*_2_), 2.74 (dd, *J* = 16.0/3.6 Hz, 1H, 3/5-C*H*_2_), 2.79–2.87 (m, 2 × 0.9H, 3/5-C*H*_2_), 4.09–4.24 (m, 4H, 2 × OC*H*_2_CH_3_), 4.60 (dd, *J* = 3.5/3.0 Hz, 1H, 2/6-C*H*), 4.69 (dd, *J* = 6.2/4.1 Hz, 1H, 2/6-C*H*), 4.81–4.83 (m, 2 × 0.9H, R_2_C=C*H*_2_), 4.90–4.93 (m, 2 × 0.1H, R_2_C=C*H*_2_*). Ratio of isomers is 9:1 (*trans*:*cis*). Signals for the *cis* diastereomer are marked with an asterisk (*). ^13^C NMR (151 MHz, CDCl_3_): *δ* (ppm) = 14.3 (1C, OCH_2_*C*H_3_), 14.5 (1C, OCH_2_*C*H_3_), 28.3 (3 × 0.9C, C(*C*H_3_)_3_), 28.4 (3 × 0.1C, C(*C*H_3_)_3_*), 33.25 (0.9C, C-3/5), 33.34 (0.9C, C-3/5), 33.6 (2 × 0.1C, C-3*, C-5*), 54.6 (1C, C-2/6), 55.7 (1C, C-2/6), 61.0 (2 × 0.1C, O*C*H_2_CH_3_*), 61.2 (0.9C, O*C*H_2_CH_3_), 61.3 (0.9C, O*C*H_2_CH_3_), 81.17 (0.9C, *C*(CH_3_)_3_), 81.21 (0.1C, *C*(CH_3_)_3_*) 112.3 (0.9C, R_2_C=*C*H_2_), 112.7 (0.1C, R_2_C=*C*H_2_*), 137.5 (0.9C, R_2_*C*=CH_2_), 138.1 (0.1C, R_2_*C*=CH_2_*), 155.2 (0.1C, N(*C*=O)O*), 155.5 (0.9C, N(*C*=O)O), 171.3 (2 × 0.1C, O=*C*OEt)*), 172.8 (2 × 0.9C, O=*C*OEt). Ratio of isomers is 9:1 (*trans*:*cis*). Signals for the *cis* diastereomer are marked with an asterisk (*). Purity (HPLC, method A): 94.4% (*t*_R_ = 21.5 min). IR (neat): *ṽ* (cm^−1^) = 2978 (CH_aliph_), 1740 (C=O_ester_), 1701 (C=O_carbamate_), 1655 (C=C), 1180, 1165, 1022 (C-N, C-O). Exact mass (APCI): *m/z* = 342.1913 (calcd. 342.1911 for C_17_H_28_NO_6_ [M+H]^+^).

#### 3.1.12. Synthesis of 1-*Tert*-Butyl 2,6-Diethyl *Trans*-4-Oxopiperidine-1,2,6-Tricarboxylate (**30**)

OsO_4_ (0.05 m in H_2_SO_4_, 3.60 mL, 0.18 mmol, 0.01 eq), pyridine (0.70 mL, 9.10 mmol, 0.5 eq) and NaIO_4_ (15.6 g, 72.8 mmol, 4.0 eq) were added to a solution of **24** (6.20 g, 18.2 mmol, 1.0 eq) in *t*-BuOH (80 mL) and H_2_O (120 mL) and the suspension was stirred at room temperature for 48 h. Then, the mixture was filtered and Na_2_SO_3_ (50 mL) and EtOAc (50 mL) were added to the solution. The aqueous layer was extracted with EtOAc (3 × 50 mL) and the combined organic layers were dried (Na_2_SO_4_), filtered and concentrated in vacuo. The residue was purified by flash column chromatography (ø = 8 cm, *h* = 16 cm, cHex/EtOAc 9:1 → 8:2, *V* = 80 mL). Colorless solid, mp 49–50 °C, yield 4.55 g (73%). C_16_H_25_NO_7_ (343.4 g/mol). TLC: *R*_f_ = 0.33 (cHex/EtOAc 8:2). ^1^H NMR (600 MHz, CDCl_3_): *δ* (ppm) = 1.25 (t, *J* = 7.1 Hz, 3H, OCH_2_C*H*_3_), 1.27 (t, *J* = 7.1 Hz, 3H, OCH_2_C*H*_3_), 1.45 (s, 9H, C(C*H*_3_)_3_), 2.72 (dd, *J* = 18.0/1.4 Hz, 1H, 3/5-C*H*_eq_), 2.86 (dd, *J* = 18.0/1.4 Hz, 1H, 3/5-C*H*_eq_), 3.01–3.08 (m, 2H, 3-C*H*_ax_, 5-C*H*_ax_), 4.12–4.26 (m, 4H, 2 × O*C*H_2_CH_3_), 4.83 (d, *J* = 7.8 Hz, 1H, 2/6-C*H*), 5.06 (d, *J* = 7.8 Hz, 1H, 2/6-C*H*). ^13^C NMR (151 MHz, CDCl_3_): *δ* (ppm) = 14.2 (1C, OCH_2_*C*H_3_), 14.3 (1C, OCH_2_*C*H_3_), 28.3 (3C, C(*C*H*_3_*)_3_), 40.6 (1C, C-3/5), 41.0 (1C, C-3/5), 53.0 (1C, C-2/6), 54.2 (1C, C-2/6), 61.9 (1C, O*C*H_2_CH_3_), 62.1 (1C, O*C*H_2_CH_3_), 81.9 (1C, *C*(CH_3_)_3_), 154.5 (1C, N(*C*=O)O), 172.3 (1C, O=*C*OEt), 172.4 (1C,O=*C*OEt), 203.8 (1C, C-4). IR (neat): *ṽ* (cm^−1^) = 2978 (CH_aliph_), 1736 (C=O_ester_), 1701 (C=O_ketone, carbamate_), 1188, 1115, 1026 (C-N, C-O). Exact mass (APCI): *m/z* = 344.1690 (calcd. 344.1704 for C_16_H_26_NO_7_^+^ [M+H]^+^).

#### 3.1.13. Synthesis of 1-*Tert*-Butyl 2,6-Diethyl *Trans*-(Ethoxycarbonylmethylene)-Piperidine-1,2,6-Tricarboxylate (**31**)

4-(Ethoxycarbonylmethylene)triphenylphosphorane (7.60 g, 21.7 mmol, 1.75 eq) was added to a solution of **30** (4.30 g, 12.4 mmol, 1.0 eq) in toluene (50 mL) and the mixture was heated at reflux for 48 h. After concentrating the mixture in vacuo, the residue was purified by flash column chromatography (ø = 6 cm, *h* = 17 cm, cHex/EtOAc 9:1, *V* = 80 mL). Colorless oil, yield 3.9 g (75%). C_20_H_31_NO_8_ (413.5 g/mol). TLC: *R*_f_ = 0.35 (cHex/EtOAc 8:2). ^1^H NMR (600 MHz, DMSO-*d*_6_): *δ* (ppm) = 1.11–1.22 (m, 9H, 3 × OCH_2_C*H*_3_), 1.35 (s, 9 × 0.54H, C(C*H*_3_)_3_), 1.36 (s, 9 × 0.46H, C(C*H*_3_)_3_*), 2.71 (dd, *J* = 16.8/2.2 Hz, 0.54H, 3/5-C*H*_2_), 2.78 (dd, *J* = 17.1/3.2 Hz, 0.46H, 3/5-C*H*_2_*), 2.83–2.95 (m, 1.46H, 3/5-C*H*_2_, 3-C*H*_2_*, 5-C*H*_2_*), 3.01 (ddm, *J* = 18.9/7.3 Hz, 0.54H, 3/5-C*H*_2_), 3.61 (dm, *J* = 18.9 Hz, 0.54H, 3/5-C*H*_2_), 3.68 (dm, *J* = 18.5 Hz, 0.46H, 3/5-C*H*_2_*), 4.02–4.17 (m, 6H, 3 × O*CH*_2_*CH_3_*), 4.57 (dd, *J* = 6.7/2.3 Hz, 0.54H, 2/6-C*H*), 4.62 (dd, *J* = 6.4/3.1 Hz, 0.46H, 2/6-C*H**), 4.67 (dd, *J* = 7.2/1.9 Hz, 0.46H, 2/6-C*H**), 4.75 (dd, *J* = 7.3/2.3 Hz, 0.54H, 2/6-C*H*), 5.80 (s, 0.54H, R_2_C=C*H*), 5.81 (s, 0.46H, R_2_C=C*H**). Ratio of rotamers is 54:46. Signals for the minor rotamer are marked with an asterisk (*). ^13^C NMR (151 MHz, DMSO-*d*_6_): *δ* (ppm) = 13.9 (0.54C, OCH_2_*C*H_3_), 14.0 (0.46C, OCH_2_*C*H_3_*), 14.0 (0.46C, OCH_2_*C*H_3_*), 14.1 (0.54C, OCH_2_*C*H_3_), 14.11 (1C, OCH_2_*C*H_3_), 27.7 (s, 3C, C(*C*H_3_)_3_), 29.8 (0.46C, C-3/5*), 30.4 (0.54C, C-3/5), 33.8 (0.54C, C-3/5), 34.0 (0.46C, C-3/5*), 51.6 (0.54C, C-2/6), 53.0 (0.46C, C-2/6*), 53.1 (0.46C, C-2/6*), 54.1 (0.54C, C-2/6), 59.5 (1C, O*C*H_2_CH_3_), 60.8 (0.46C, O*C*H_2_CH_3_*), 60.9 (0.54C, O*C*H_2_CH_3_), 61.0 (0.46C, O*C*H_2_CH_3_*), 61.1 (0.54C, O*C*H_2_CH_3_), 80.3 (0.54C, C(CH_3_)_3_), 80.4 (0.46C, *C*(CH_3_)_3_*), 116.9 (0.54C, R_2_C=*C*H_2_), 117.0 (0.46C, R_2_C=*C*H_2_*), 151.4 (0.46C, R_2_*C*=CH_2_*), 151.6 (0.54C, R_2_*C*=CH_2_), 154.0 (0.54C, N(*C*=O)O), 154.2 (0.46C, N(*C*=O)O*), 165.0 (1C, O=*C*OEt), 171.5 (0.46C, O=*C*OEt*), 171.7 (0.54C, O=*C*OEt), 172.0 (0.54C, O=*C*OEt), 172.2 (0.46C, O=*C*OEt*). Ratio of rotamers is 54:46. Signals for the minor rotamer are marked with an asterisk (*). Purity (HPLC, method A): 98.9% (*t*_R_ = 22.1 min). IR (neat): *ṽ* (cm^−1^) = 2978 (C-H_aliph_), 1744 (C=O_ester_), 1701 (C=O_carbamate_), 1651 (C=C), 1184, 1142, 1026 (C-N, C-O). Exact mass (APCI): *m/z* = 414.2144 (calcd. 414.2122 for C_20_H_32_NO_8_^+^ [M+H]^+^).

#### 3.1.14. Synthesis of Diethyl *Cis*/*Trans*-4-Methylenepiperidine-2,6-Dicarboxylate (**32**)

TFA (6.5 mL, 87.9 mmol, 30 eq) was added to a solution of **24** (1.0 g, 2.9 mmol, 1.0 eq) in dry CH_2_Cl_2_ (50 mL) and the mixture was stirred at room temperature for 16 h. The next day, Na_2_CO_3_ was added, the layers were separated and the aqueous layer was extracted with CH_2_Cl_2_ (2 × 40 mL). The combined organic layers were dried (Na_2_SO_4_), filtered and concentrated in vacuo. The diastereomers were separated twice by flash column chromatography (1.50 g cartridge, cHex/EtOAc 8:2 → 6:4; 2. 25 g cartridge, cHex/EtOAc 8:2). C_12_H_19_NO_4_ (241.3 g/mol). *cis*-**32**: Yellow oil, yield 79 mg (11%). TLC: *R*_f_ = 0.31 (cHex/EtOAc 1:1). ^1^H NMR (600 MHz CDCl_3_) *δ* = 1.29 (t, *J* = 7.1 Hz, 6H, 2 × OCH_2_C*H*_3_), 2.10–2.17 (m, 2H, 3-C*H*_ax_, 5-C*H*_ax_), 2.62 (dd, *J* = 13.5/2.7 Hz, 2H, 3-C*H*_eq_, 5-C*H*_eq_), 3.37 (dd, *J* = 11.8/3.0 Hz, 2H, 2-C*H*_ax_, 6-C*H*_ax_), 4.22 (dq, *J* = 7.1/1.4 Hz, 4H, 2 × OC*H*_2_CH_3_), 4.87 (t, *J* = 1.7 Hz, 2H, R_2_C=C*H*_2_). N*H* signal is missing. ^13^C NMR (151 MHz, CDCl_3_) *δ* = 14.3 (2C, 2 × OCH_2_*C*H_3_), 37.7 (2C, C-3, C-5), 58.9 (2C, C-2, C-6), 61.4 (2C, 2 × O*C*H_2_CH_3_), 111.4 (1C, R_2_C=*C*H_2_), 142.5 (1C, C-4), 171.9 (2C, 2 × O=*C*OEt)). IR (neat): *ṽ* (cm^−1^) = 2982 (C-H_aliph_), 1732 (C=O_ester_), 1651 (C=C), 1180, 1026 (C-N, C-O). Exact mass (APCI): *m/z* = 242.1360 (calcd. 242.1387 for C_12_H_20_NO_4_ [M+H]^+^). *trans*-**32**: Yellow oil, yield 623 mg (89%). TLC: *R*_f_ = 0.20 (cHex/EtOAc 1:1). ^1^H NMR (600 MHz, CDCl_3_): *δ* (ppm) = 1.29 (t, *J* = 7.1 Hz, 6H, 2 × OCH_2_C*H*_3_), 2.46 (dd, *J* = 13.2/7.0 Hz, 2H, 3-C*H*_2_, 5-C*H*_2_), 2.56 (dd, *J* = 13.2/5.0 Hz, 2H, 3-CH_2_, 5-CH_2_), 3.84 (dd, *J* = 7.0/5.0 Hz, 2H, 2-C*H*, 6-C*H*), 4.14–4.24 (m, 4H, 2 × OC*H*_2_CH_3_), 4.87 (s, 2H, R_2_C=C*H*_2_). N*H* signal is missing. ^13^C NMR (151 MHz, CDCl_3_): *δ* (ppm) = 14.4 (2C, 2 × OCH_2_*C*H_3_), 36.3 (2C, C-3, C-5), 56.1 (2C, C-2, C-6), 61.2 (2C, 2 × O*C*H_2_CH_3_), 111.7 (1C, R_2_C=*C*H_2_), 141.3 (1C, C-4), 172.7 (2C, 2 × O=*C*OEt). IR (neat): *ṽ* (cm^−1^) = 3356 (N-H), 2978 (C-H_alip_), 1728 (C=O_ester_), 1655 (C=C), 1200, 1165, 1026 (C-N, C-O). Exact mass (APCI): *m/z* = 242.1373 (calcd. 242.1387 for C_12_H_20_NO_4_^+^ [M+H]^+^).

#### 3.1.15. Synthesis of Diethyl *Trans*-4-Oxopiperidine-2,6-Dicarboxylate (**33**)

TFA (3.60 mL, 47.0 mmol, 30 eq) was added to a solution of **30** (0.54 g, 1.57 mmol, 1 eq) in CH_2_Cl_2_ (15 mL) and the mixture was stirred at room temperature for 16 h. Then, the mixture was washed with NaHCO_3_ (30 mL). The aqueous phase was extracted with CH_2_Cl_2_ (3 × 20 mL) and the combined organic layers were dried (Na_2_SO_4_), filtered and concentrated in vacuo. Yellow oil, yield 0.27 g (71%). C_11_H_17_NO_5_ (243.3 g/mol). TLC: *R*_f_ = 0.22 (cHex/EtOAc 1:1). ^1^H NMR (600 MHz, CDCl_3_): *δ* (ppm) = 1.28 (t, *J* = 7.1 Hz, 6H, 2 × OCH_2_C*H*_3_), 2.61 (ddd, *J* = 15.1/7.0/1.4 Hz, 2H, 3-C*H*_ax_, 5-C*H*_ax_), 2.71 (ddd, *J* = 15.0/5.5/1.3 Hz, 2H, 3-C*H*_eq_, 5-C*H*_eq_), 4.05 (dd, *J* = 7.0/5.5 Hz, 2H, 2-CH, 6-CH), 4.21 (q, *J* = 7.1 Hz, 4H, 2 × OC*H*_2_CH_3_). The signal for N*H* is not observed in the spectrum. ^13^C NMR (151 MHz, CDCl_3_): *δ* (ppm) = 14.3 (2C, 2 × OCH_2_*C*H_3_), 42.7 (2C, C-3, C-5), 54.8 (2C, C-2, C-6), 61.8 (2C, 2 × O*C*H_2_CH_3_), 171.7 (2C, 2 × O=*C*OEt), 204.7 (1C, C-4).

#### 3.1.16. Synthesis of Diethyl *Trans*-4-(2-Ethoxy-2-Oxoethylidene)Piperidine-2,6-Dicarboxylate (**34**)

TFA (2.10 mL, 28.4 mmol, 30 eq) was added to a solution of **31** (0.39 g, 0.95 mmol, 1 eq) in CH_2_Cl_2_ (10 mL) and the mixture was stirred at room temperature for 16 h. Then, the mixture was washed with NaHCO_3_ (20 mL). The aqueous phase was extracted with CH_2_Cl_2_ (3 × 15 mL) and the combined organic layers were dried (Na_2_SO_4_), filtered and concentrated in vacuo. Yellow oil, yield 0.27 g (90%). C_15_H_23_NO_6_ (313.4 g/mol). TLC: *R*_f_ = 0.35 (cHex/EtOAc 1:1). ^1^H NMR (600 MHz, CDCl_3_): *δ* (ppm) = 1.25 (t, *J* = 7.1 Hz, 3H, OCH_2_C*H*_3_), 1.27 (t, *J* = 7.1 Hz, 3H, OCH_2_C*H*_3_), 1.28 (t, *J* = 7.1 Hz, 3H, OCH_2_C*H*_3_), 2.52 (dd, *J* = 13.3/7.1 Hz, 1H, 3/5-C*H*_ax_), 2.61 (dd, *J* = 13.3,/5.0 Hz, 1H, 3/5-C*H*_eq_), 3.21 (dd, *J* = 13.8, 6.9 Hz, 1H, 3/5-C*H*_ax_), 3.26 (dd, *J* = 13.8/5.3 Hz, 1H, 3/5-C*H*_eq_), 3.86 (dd, *J* = 6.9/5.3 Hz, 1H, 2/6-C*H*), 3.92 (dd, *J* = 7.1/5.0 Hz, 1H, 2/6-C*H*), 4.12–4.24 (m, 6H, 3 × OC*H*_2_CH_3_), 5.75 (s, 1H, R_2_C=C*H*). ^13^C NMR (151 MHz, CDCl_3_): *δ* (ppm) = 14.3 (1C, OCH_2_*C*H_3_), 14.4 (2C, 2 × OCH_2_*C*H_3_), 31.2 (1C, C-3/5), 38.2 (1C, C-3/5), 55.6 (1C, C-2/6), 56.0 (1C, C-2/6), 60.0 (1C, O*C*H_2_CH_3_), 61.3 (1C, O*C*H_2_CH_3_), 61.4 (1C, O*C*H_2_CH_3_), 117.5 (1C, R_2_C=*C*H), 153.4 (1C, R_2_*C*=CH), 166.0 (1C, O=*C*OEt), 172.2 (1C, O=*C*OEt), 172.6. (1C, O=*C*OEt).

#### 3.1.17. Synthesis of Diethyl 4-Methylpyridine-2,6-Dicarboxylate (**38**)

At 0 °C, NaOCl (0.75 m in H_2_O, 3.90 mL, 2.90 mmol, 2 eq) was added to a solution of **32** (350 mg, 1.45 mmol, 1 eq) and AcOH (0.17 mL, 2.90 mmol, 2 eq) in *t*-BuOH (0.20 mL, 1.74 mmol, 1.2 eq) and methyl *t*-butyl ether (8 mL) and the mixture was stirred for 1 h. Then, DIPEA (2.05 mL, 11.6 mmol, 8 eq) was added and the mixture was stirred at room temperature for 16 h. After the addition of H_2_O (10 mL), the mixture was extracted with EtOAc (3 × 15 mL). The combined organic layers were dried (Na_2_SO_4_), filtered and concentrated in vacuo. The residue was purified by flash column chromatography (25 g cartridge, cHex/EtOAc 95:5 → 9:1). Yellow oil, yield 20 mg (6%). C_12_H_15_NO_4_ (237.3 g/mol). TLC: *R*_f_ = 0.59 (cHex/EtOAc 9:1). ^1^H NMR (600 MHz, CDCl_3_): *δ* (ppm) = 1.45 (t, *J* = 7.2 Hz, 6H, 2 × OCH_2_C*H*_3_), 2.51 (s, 3H, C*H*_3_), 4.47 (q, *J* = Hz, 4H, 2 × OC*H*_2_CH_3_), 8.10 (s, 2H, 3-CH_arom_, 5-CH_arom_). ^13^C NMR (151 MHz, CDCl_3_): *δ* (ppm) = 14.4 (2C, 2 × OCH_2_*C*H_3_), 21.3 (1C, *C*H_3_), 62.4 (2C, 2 × O*C*H_2_CH_3_), 128.8 (2C, C-3_arom_, C-5_arom_), 148.6 (2C, C-2_arom_, C-6_arom_), 150.2 (1C, C-4_arom_), 165.1 (2C, 2 × O=*C*OEt).

#### 3.1.18. Synthesis of Diethyl (RS)-4-Oxo-1,2,3,4-Tetrahydropyridine-2,6-Dicarboxylate (**36**)

At 0 °C, NaOCl (0.75 m in H_2_O, 1.70 mL, 1.28 mmol, 1.2 eq) was added to a solution of **33** (260 mg, 1.07 mmol, 1 eq) and AcOH (0.07 mL, 1.28 mmol, 1.2 eq) in *t*-BuOH (0.12 mL, 1.28 mmol, 1.2 eq) and methyl *t*-butyl ether (10 mL) and the mixture was stirred for 1 h. Then, DIPEA (0.90 mL, 5.35 mmol, 5 eq) was added and the mixture was stirred at room temperature for 16 h. H_2_O (15 mL) was added and the mixture was extracted with EtOAc (3 × 15 mL) and dried (Na_2_SO_4_), filtered and concentrated in vacuo. The residue was purified by flash column chromatography (25 g cartridge, cHex/EtOAc 7:3 → 1:1). Colorless oil, yield 101 mg (39%). C_11_H_15_NO_5_ (241.2 g/mol). TLC: *R*_f_ = 0.40 (cHex/EtOAc 1:1). ^1^H NMR (600 MHz, CDCl_3_): *δ* (ppm) = 1.30 (t, *J* = 7.1 Hz, 3H, OCH_2_C*H*_3_), 1.35 (t, *J* = 7.1 Hz, 3H, OCH_2_C*H*_3_), 2.70 (dd, *J* = 16.5/12.3 Hz, 1H, 3-C*H*_2_), 2.80 (dd *J* = 16.5/5.8 Hz, 1H, 3-C*H*_2_), 4.23–4.29 (m, 2H, OC*H*_2_CH_3_), 4.32–4.37 (m, 3H, 2-C*H*, OC*H*_2_CH_3_), 5.77 (s, 1H, 5-C*H*), 6.07 (s, 1H, N*H*). ^13^C NMR (151 MHz, CDCl_3_): *δ* (ppm) = 14.1 (1C, OCH_2_*C*H_3_), 14.2 (1C, OCH_2_*C*H_3_), 38.2 (1C, C-3), 54.8 (1C, C-2), 62.4 (1C, O*C*H_2_CH_3_), 63.0 (1C, O*C*H_2_CH_3_), 102.2 (1C, C-5), 147.7 (1C, C-6), 162.9 (1C, O=*C*OEt), 169.9 (1C, O=*C*OEt), 192.5 (1C, C-4).

#### 3.1.19. Synthesis of Diethyl (*E*)- and (*Z*)-(RS)-4-(2-Ethoxy-2-Oxoethylidene)-1,2,3,4-Tetrahydropyridine-2,6-Dicarboxylate (**37**)

At 0 °C, NaOCl (0.75 m in H_2_O, 2.20 mL, 1.65 mmol, 2 eq) was added to a solution of **34** (260 mg, 0.82 mmol, 1 eq) and AcOH (0.10 mL, 1.65 mmol, 2 eq) in *t*-BuOH (0.10 mL, 1.00 mmol, 1.2 eq) and methyl *t*-butyl ether (8 mL) and the mixture was stirred for 1 h. Then, DIPEA (1.15 mL, 6.61 mmol, 8 eq) was added and the mixture was stirred at room temperature for 16 h. H_2_O (10 mL) was added and the mixture was extracted with EtOAc (3 × 15 mL) and dried (Na_2_SO_4_), filtered and concentrated in vacuo. The residue was purified by flash column chromatography (25 g cartridge, cHex/EtOAc 8:2 → 6:4). Yellow oil, yield 77 mg (30%). C_15_H_21_NO_6_ (311.3 g/mol). TLC: *R*_f_ = 0.49 (cHex/EtOAc 7:3). ^1^H NMR (600 MHz, CDCl_3_): *δ* (ppm) = 1.24–1.32 (m, 6H, 2 × OCH_2_C*H*_3_, 2 × OCH_2_C*H*_3_*), 1.34 (t, *J* = 7.1 Hz, 3 × 0.55H, OCH_2_C*H*_3_), 1.35 (t, *J* = 7.1 Hz, 3 × 0.45H, OCH_2_C*H*_2_*), 2.76 (ddd, *J* = 15.5/9.6/1.5 Hz, 0.45H, 3-C*H*_2_*), 2.83 (ddd, *J* = 15.5/5.0/1.2 Hz, 0.45H, 3-C*H*_2_*), 3.10 (ddd, *J* = 17.0/10.2/2.0 Hz, 0.55H, 3-C*H*_2_), 3.70 (ddd, *J* = 17.0/4.8/1.6 Hz, 0.55H, 3-C*H*_2_), 4.00 (dd, *J* = 10.2/4.8 Hz, 0.55H, 2-C*H*), 4.06 (dd, *J* = 9.6/5.0 Hz, 0.45H, 2-C*H**), 4.12–4.27 (m, 4H, 2 × OC*H*_2_CH_3_, 2 × OC*H*_2_CH_3_*), 4.30 (q, *J* = 7.1 Hz, 1.10H, OC*H*_2_CH_3_), 4.31 (q, *J* = 7.1 Hz, 0.9H, OC*H*_2_CH_3_*), 5.43 (s, 0.45H, 5-C*H**), 5.66 (s, 0.55H, 5-C*H*), 6.08 (s, 0.55H, C*H*COOEt), 7.37 (s, 0.45H, C*H*COOEt*). Ratio of diastereomers is 55:45. The signals for the minor diastereomer are marked with an asterisk (*). ^13^C NMR (151 MHz, CDCl_3_): *δ* (ppm) = 14.25, 14.28, 14.20, 14.31 (2C, 2 × OCH_2_*C*H_3_, 2 × OCH_2_*C*H_3_*), 14.50, 14.51 (1C, OCH_2_*C*H_3_, OCH_2_*C*H_3_*), 28.4 (0.55C, 3-C), 33.7 (0.45C, C-3*), 53.5 (0.55C, C-2), 55.8 (0.45C, C-2*), 59.82 (0.55, O*C*H_2_CH_3_), 59.84 (0.45C, O*C*H_2_CH_3_*), 61.8 (0.45C, O*C*H_2_CH_3_*), 61.9 (0.55C, O*C*H_2_CH_3_), 62.06 (0.55C, O*C*H_2_CH_3_), 62.13 (0.45C, O*C*H_2_CH_3_*), 102.4 (0.45C, *C*HCOOEt*), 107.1 (0.55C, *C*HCOOEt), 112.3 (0.45C, C-5*), 113.2 (0.55C, C-5), 137.9 (0.55C, C-6), 138.1 (0.45C, C-6*), 146.1 (0.45C, C-4*), 148.0 (0.55C, C-4), 163.2 (0.55C, O=*C*OEt), 163.9 (0.45C, O=*C*OEt*), 166.7 (0.45C, O=*C*OEt*), 167.0 (0.55C, O=*C*OEt), 170.8 (0.45C, O=*C*OEt*), 171.2 (0.55C, O=*C*OEt). Ratio of diastereomers is 55:45. The signals for the minor diastereomer are marked with an asterisk (*).

## 4. Conclusions

In order to learn more about the relevance of the carboxy moieties of betalamic acid (**9**), the seven-step synthesis of the betalamic acid analog **13** without carboxy groups in positions C-2-and C-6 was designed and carried out. Due to low stability, in particular against O_2_, the free amine **13** could be characterized only by NMR spectroscopy. However, the Boc-protected precursor **21** could be isolated. In the Folin–Ciocalteu assay, **21** did not show any antioxidative properties, indicating that a free amine within the piperidine ring is essential for its antioxidative activity. Analogous 1,2,3,4-tetrahydropyridines **35–37** with two ester moieties in positions C-2 and C-6 and different substituents in position C-4 showed different levels of stability, i.e., different antioxidative properties in NMR studies.

## Data Availability

Sample of the compounds and all data are available from the authors.

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
