# Peer review of "Synthesis and Antioxidative Properties of 1,2,3,4-Tetrahydropyridine Derivatives with Different Substituents in 4-Position"

_molecules, 2022, doi:10.3390/molecules27217423_

Round 1
Reviewer 1 Report
Dear Authors
The background and the introduction to the subject are well presented, the suggested synthesis plan slightly complicated which led to an unstable product. They were unable to evaluate the antioxidant activity of the targeted product (13) due to its instability, hence they tested the antioxidant activity of the protected compound (21) which gave a negative result, based on that, they assumed 13 would have some antioxidant activity! I think section 2.2 ‘’Antioxidant activity …’’ must be improved or corrected. Also in the same section, it is unclear how the used the NMR to evaluate the antioxidant activities of the compounds 35 to 37, the part needs some improvement too.
Best regards,
Reviewer 2 Report
First of all, I would like to emphasize that the authors presented very significant results in this field of chemistry. Also, they wrote this paper very well.
Comment 1. Considering the results presented in the paper, the abstract should be improved, insert more results into the abstract.
Comment 2. Some of keywords should be replaced with more specific keywords to the article.
Comment 3. Page 3 (at the end) – You use CH3OH and EtOH. It must be uniform (CH3OH and CH3CH2OH or MeOH and EtOH).
Comment 4. Bold all labels of compounds in the text.
Reviewer 3 Report
In this paper, Diana and co-workers describe two synthetic strategies towards piperidine derivatives, analogues of betalamic acid. The synthetic procedures seems to be carefully described and provided with the proper spectroscopic description. In the experimental procedures authors included even Rf‘s of the products which is a nice point, very often omitted by other research groups.
However, some concerns arose during the studying of the presented paper.
Major concerns:
1) Since the main scientific emphasis of the paper has been put on the synthetic aspect of the research, the introduction part should more thoroughly discuss the synthetic context of the presented research. It does not seem as an important figure to discuss biosynthetic pathways towards betalains. While, it would be of the high value if authors compare their synthetic strategies with known strategies described by other research groups. What makes their work different from other chemists dealing with analogues of betalamic acid? What made them to study compounds 21, 35-37 as final synthetic targets? I believe this answers if included in the introduction, it would help readers to appreciate the scientific sound of the paper.
2) Authors very often declare necessity of dealing with stereoisomers which is a very common problem in organic chemistry research. I can understand that it is the reason why many spectra seems to be impure. However, in some cases even spectra of single isomers seems to have some unknown signals or general impurities too (compounds 25, 28, 37, 38). Furthermore, the quality of 1H NMR of 17 is concerning, the residual signal is not of the proper multiplicity suggesting bad spectrum shimming, on 13C NMR the signal/noise ratio is not high enough to interpret the spectrum. If the research is mostly dedicated to organic synthesis, the spectra should be corrected.
3) What made authors to conclude the geometry of the double bond on isomer 16 as an E- isomer? It is a not trivial spectroscopic problem to solve and the discussion of it would be valuable in the manuscript.
4) If authors used only one isomer to the synthesis of 15 from 16 how they explain formation of two diastereomers of 15 since the reaction took place quite far from the double bond that has been determined as E- isomer at the previous step?
Minor concerns:
5) The compounds numbering does not seem to be very reader-friendly. For example, the products of progressive step of the syntheses have lower numbers. Moreover, the spectra in the supplementary materials does not appear in the order, it makes it more difficult to the reader to find the spectrum of interest. Similar problem I encountered while I studied experimental procedures.
6) Some subscripts are missing at the page 2 of SM.
7) The name of compounds in the experimental part should be edited. Some expressions such as cis-, trans-, tert-, E-, Z- should be written in italic which is not the case in most of the names.
8) The authors very often call positions in piperidine derivatives as positions 2- and 6-. It is a kind of scientific shorthand not fully adequate for scientific papers. The more correct form of it should be positions C-2 and C-6.
9) I am concern about essentiality of discussion of biosynthetic pathways, however, if authors decide to keep it in the article, in the words like L-DOPA and L-tyrosine, the letter l should be written with a small cap not with capital letters.
10) The authors discuss possible relative configuration of substituents on 24, I do agree with their conclusions in general. However, while it is justified in cis- isomer to call signal of exocyclic methylene protons as singlet, the trans- isomer does not result in a formation of singlet since these protons are not chemically equivalent. Signals might be coincidentally superimposed, however, it should be described as multiplet or singlet-like multiplet if so.
11) On Scheme 6 point a), the conditions description might be misleading. It suggests that butyllithium and isopropylamine were in contact with 25. I suggest changing it onto “LDA” which was a real reactive species. The exact procedure how LDA was generated is already described in the experimental section which is a good decision.
12) The Scheme 7 does not look fully discussed in the main text, it might be an editorial mistake. I do not see any discussion of compounds 38, 39, 40 in text close to the Scheme 7. There is no procedure of 35 in the experimental section.
13) Authors state that compound 32 was quickly oxidised on the contrary to some other derivatives. When I read this statement I was curious how quick was this process. I believe, it would be an interesting information for readers too.
14) Some references should be rechecked. The reference 45 has some mistakes in the title. On the other hand the reference of 38 and 40 does not include the scientific medium that authors want to cite.
15) At page 8 paragraph 1 authors write: In this group of assays, an oxidant is necessary to evaluate the ability of the tested antioxidant to reduce it.
I am not fully convinced if the sentence reflects the real meaning. I suggest rephrasing.
Reviewer 4 Report
The manuscript by Diana et al. well introduced the importance and attractive property of the betalains as one of the major color sources of fruits. The retosynthetic analysis is also thoroughly discussed and proved to be an accessible route to provide the desired derivatives. The antioxidative properties of the final analogues were carefully characterized. Most of the results are consistent yet there are limitations that warrant attention:
1. “DOPA” is not defined in the introduction part.
2. As compound 24 was obtained as mixture of trans- and cis isomers (9:1), it might be more consistent to emphasize the result of compound 30. Is it still a mixture or a pure diastereomer?
3. Following point 2, if 30 was a mixture, compound 31 is also expected to be a mixture of four isomers (two from cis-30 and two from trans-30) instead of two rotamers.
4. Could you also provide the more details about the two rotamers of 31 as stated in the experimental section?
5. Scheme 7 contains much more information than the paragraph “Since the piperidines 24, 30, and 31 …”. It might be a little bit confusing at first glance since the synthesis of 35-37 were described in third paragraph of the next section.
6. In Scheme 7, consider showing the arrow from 36 to 39 as “no-go”?
7. In the legend of scheme 7, yield should be 30% (37) for condition (b).
8. Something went wrong for the format of “3H2O-P2O5-13WO3-5MoO3-10H2O”
9. In figure 3, there should be no wadged-bond drawing for compound 37.
10. One might be curious about whether the two isomers of 37 shows different oxidation rate against oxygen. This could be revealed by comparing the remaining peak of their distinctive 1H NMR spectrum.
11. For synthesis of 18, is there a specific reason to state the final ration of petroleum ether/EtOAc as 4/4 instead of either 1/1 or 5/5(keeping the sum as 10)?
12. There are some extra or missing “,” in the experimental section, For example: the 1H NMR for compound 15.
13. There are some H/C are formatted as italic unnecessarily. For example: the 1H NMR for compound 17, 4.78 (s, 1H, 3-CH); 13; 24; 31; etc.
14. Carbon peaks are assigned in a different format for compound 20: “3-CH” instead of “C-3”.
15. There’s an extra row at the end of references.

Round 2
Reviewer 3 Report
.
